# Natural Plant Extracts: An Update about Novel Spraying as an Alternative of Chemical Pesticides to Extend the Postharvest Shelf Life of Fruits and Vegetables

**DOI:** 10.3390/molecules27165152

**Published:** 2022-08-12

**Authors:** Muhammad Umar Shahbaz, Mehwish Arshad, Kinza Mukhtar, Brera Ghulam Nabi, Gulden Goksen, Małgorzata Starowicz, Asad Nawaz, Ishtiaq Ahmad, Noman Walayat, Muhammad Faisal Manzoor, Rana Muhammad Aadil

**Affiliations:** 1Plant Pathology Research Institute, AARI, Faisalabad 38000, Pakistan; 2National Institute of Food Science and Technology, University of Agriculture, Faisalabad 38000, Pakistan; 3Department of Food Technology, Vocational School of Technical Sciences at Mersin Tarsus Organized Industrial Zone, Tarsus University, Mersin 33100, Turkey; 4Department of Chemistry and Biodynamics of Food, Institute of Animal Reproduction and Food Research, 10-784 Olsztyn, Poland; 5Shenzhen Key Laboratory of Marine Microbiome Engineering, Institute for Advanced Study, Shenzhen University, Shenzhen 518060, China; 6College of Food Science and Technology, Zhejiang University of Technology, Hangzhou 310014, China; 7School of Food Science and Engineering, South China University of Technology, Guangzhou 510641, China; 8Guangdong Provincial Key Laboratory of Intelligent Food Manufacturing, Foshan University, Foshan 528000, China

**Keywords:** human health, pesticide utilization, postharvest quality, shelf life

## Abstract

Fresh fruits and vegetables, being the source of important vitamins, minerals, and other plant chemicals, are of boundless importance these days. Although in agriculture, the green revolution was a milestone, it was accompanied by the intensive utilization of chemical pesticides. However, chemical pesticides have hazardous effects on human health and the environment. Therefore, increasingly stimulating toward more eco-friendly and safer alternatives to prevent postharvest losses and lead to improving the shelf life of fresh fruits and vegetables. Proposed alternatives, natural plant extracts, are very promising due to their high efficacy. The plant-based extract is from a natural source and has no or few health concerns. Many researchers have elaborated on the harmful effects of synthetic chemicals on human life. People are now much more aware of safety and health concerns than ever before. In the present review, we discussed the latest research on natural alternatives for chemical synthetic pesticides. Considering that the use of plant-based extracts from aloe vera, lemongrass, or neem is non-chemical by-products of the fruits and vegetable industry, they are proved safe for human health and may be integrated with economic strategies. Such natural plant extracts can be a good alternative to chemical pesticides and preservatives.

## 1. Introduction

Fresh fruits and vegetables keep the activities of natural biological processes even after harvest from parental plants. Such a process could bring about several undesirable changes that ultimately contribute to postharvest quality because of shriveling, mass loss, poor consumption quality, and short storage life [1,2]. These are the significant limiting restrictions defining the keenness of fresh fruits and vegetables in the marketplace [3]. The ripening is a continuous process of ripening in many fruits and vegetables even after harvest. It is a highly synchronized evolving progression that conveys by a series of biochemical changes such as color changes, reduction of astringency, volatile production, tissue softening, seed maturation, taste, and many others, leading to the quality deterioration [4]. Postharvest diseases are the main factors that contribute to quality losses to the produce. Additionally, wrong transportation and preservation can influence fresh produces. After harvesting, the product can be more susceptible to numerous postharvest ailments and diseases instigated by bacterial and fungal pathogens [5]. Approximately 40% of fruit and vegetable losses were reported throughout the postharvest handling and supply chain [6].

For many years, farmers preserved fruits and vegetables using synthetic chemical preservatives. These chemicals, on the one hand, increase the shelf life and maintain the postharvest quality of fruits and vegetables. On the other hand, such chemicals bring potentially dangerous compounds into the food chain and cause harmful effects on consumers’ health [7,8]. Continuously, postharvest disease management has been conducted by using chemicals, but improper use of such chemicals can ground numerous unwanted penalties [9]. The fast decline in the quality attributes of fresh fruits and vegetables during the postharvest period is a problematic issue in the present times.

Pesticide is a multifaceted word that is used collectively for all chemicals that are applied to terminate pests; this includes insecticides, fungicides, herbicides, etc. [10]. These are the chemicals that farmers use in agricultural land, gardens, and other areas to destroy pests and microorganisms, which are undesirable. In addition to being effective against harmful pests and microorganisms, these pesticides in higher concentrations also can cause harm to humans. A report by the World Health Organization (WHO) explored that about 1 million humans are directly being affected by acute poisoning caused by the abused use of pesticides (death rate of 0.4–1.9% every year) [11]. Exported fruits are exposed to a prolonged handling and supply chain, so often follow-on loss in terms of quality such as decay, shrivel, over-ripeness, and weight loss. The industry of fruits and vegetables strongly relies on the application of synthetic chemicals such as chlorine dioxide, nitric oxide, salicylic acid, and 1-methyl cyclopropane as a postharvest treatment for maximizing the economic potential of their products [12,13]. Though such chemical treatments show potential effects in maintaining the quality and prolonging the shelf life of fruits and vegetables, there are still many disadvantages to their application [14]. The use of chemical agents on fruits seems to bias objectionable when it comes to consumer preference. Nowadays, consumers are seeking safer and healthier foods with the lowest additives or synthetic agent addition [15]. Chlorpyrifos is one of the pesticides for fruits and vegetables that are showing up. That is nerve poisoning organophosphorus insecticide, which is used for conventionally produced citrus fruits. Chlorpyrifos is a hazardous material for humans and the environment.

Pesticide poisoning is a major public health problem and accounts for nearly 300,000 deaths per year worldwide. The human body exposed to pesticides has more DNA damage even if there is no detectable quantity of pesticides in the body. Their metabolites present in human biological matrices not only affect the DNA but also inhibit the acetylcholinesterase (AChE) activity [16]. The oxidative stress due to pesticide exposure to the human body may lead to DNA damage, the development of Parkinson’s and Alzheimer’s, and many other disorders [17]. In contrast, there is a noteworthy sign that agricultural use of these pesticides has a foremost influence on the quality of water, and when used, they can cause severe health concerns [18]. Consumers are currently concerned about the use of fungicide-sprayed fruits, as their active compounds and co-formulants have been linked with a variety of health and environmental contamination concerns. Because the modes of action of the drug are not type-specific, concerns about the environmental risk associated with them are expressed in different ways (for example, residues in food and drinking water) over a short period (e.g., skin and eye pain, headache, dizziness, and lightheadedness) to common side effects (e.g., cancer, form, and diabetes). Their risks are difficult to explain due to exposure of different groups (e.g., duration and level of exposure, types of disinfectants, relative to toxicity and duration) and environmental characteristics of the group concerned. Therefore, the development of pesticides and other combined experimental methods (Integrated Pest Management) is required to minimize the effects of such chemical pesticides [19]. The increasing demand for safe and healthy production needs to transfer from synthetic to natural safe chemicals. Plant extracts are of significant importance in this regard [10].

Plant extract use in the world market has a significant scope. This market is growing very fast, and the growth is projected to reach USD 55.3 billion by 2026 [20]. Consumable plant-based extracts can be used on fresh products to improve and maintain their quality and increase storage life and time usability. The high demand for food products free from pesticide residues has pushed researchers to work progressively in search of environment-friendly alternatives with fewer or no health hazards. Plant-based extracts are known to be safer natural preservatives and turn out to be a steadfast substitute for hazardous synthetic chemical pesticides and preservatives, particularly for the resist postharvest diseases and maintaining postharvest quality [21]. Thus, the utilization of affordable plant-based natural preservatives has shown great reflection for preserving fruit quality and shelf life with great effectiveness. Moreover, around the world, consumer awareness is increasing concerning the quality of fruits and vegetables and their safety of high nutrition. Consequently, there is an increasing interest in replacing synthetic chemicals with natural plant extracts, for instance, aloe vera, neem, lemongrass, and several types of oligosaccharides for preserving postharvest quality and enhancing the shelf life of fruits and vegetables. In this sense, utilizing natural plant-based extracts as preservatives has paid a significant attraction in postharvest treatments of fruits and vegetables for increasing the shelf life and maintaining the nourishing components [22].

The extracts from different plants accessible worldwide at a low cost can be used to cope with losses due to postharvest diseases. Such plant extracts impose no damaging effects on the quality of the produce and public health. Extracts of plants with positive potential for antioxidant activity have been widely studied for scavenging the free radicals and strong capacity caused by the biologically active components such as punicalagin, ellagic, gallic, and chlorogenic acids [23,24,25]. Researchers examined pomegranate peel extracts as a natural inhibitor of pathogenic fungi and bacteria [23,24,25]. Plant extract application has been extensively reported to minimize postharvest spoilage, prolong shelf life, and maintain the postharvest quality of fruits and vegetables. The potential application of natural products such as antimicrobials and antioxidants on fruits and vegetables may be considered as alternatives for the conservation of postharvest quality and prolongation of their shelf life [26].

This review aims to provide a general view of the utilization of natural plant extracts as an alternative to chemical pesticides for fruits and vegetable spraying, specifically as edible spraying to protect the spoilage of fresh fruits and vegetables, the health of consumers, and the environment. An overview of the natural plant extract utilization is conducted, and the recent studies of their impact on microbiological safety, prolongation of shelf life, and nutritional and physiochemical quality of fruits and vegetables are presented.

## 2. Properties of Natural Plant Extract

Plant extract utilization has been increased as an additive in the food industry due to the presence of bioactive compounds such as polyphenols and carotenoids [27]. These natural extracts have antimicrobial and antioxidant properties [28]. Polyphenols and carotenoids prevent oxidative changes, development of off flavor, and increase color stability and shelf life of the product. These natural extracts can be utilized to replace the synthetic compounds, which are harmful and exerts carcinogenic impacts. Moreover, there is also a big challenge of efficient extraction of natural extracts, their applications in industry, and producing products with fewer health hazards. Consumers prefer high-quality products that are without chemical additives and have a high shelf life. Different types of natural edible coatings have been developed to fulfill society’s demands [29,30].

### 2.1. Aloe Vera Extract

Aloe vera is an herbal plant and has medicinal properties; hence, its applications in food, pharmaceutical and cosmetic industries have been studied [31]. It has a jelly-like texture and is composed of various bioactive compounds as well as carbohydrates, proteins, fibers, soluble sugars, vitamins, minerals, amino acids, organic acids, and phenolic compounds [32]. In functional food development, aloe vera is utilized as an edible coating film [31]. Aloe vera gel is an excellent example of active packaging due to its antimicrobial and antioxidant properties. It has been investigated as an effective preservative in terms of shelf-life extension [33]. Aloe vera gel exhibits antifungal properties to prevent postharvest diseases. Aloe vera gel has proven effective in reducing the spore survival of *Penicillium*, *Botrytis* and *Alternaria* by 15–20% and those of *Rhizoctonia*, *Fusarium,* and *Colletotrichum* by 22–38%. Aloe vera gel utilization in blueberries [34], strawberries [35], and avocado [36] as antifungal coating has been proven excellent. Additionally, aloe vera has antibacterial activities against *Bacillus cereus*, *Salmonella typhimurium*, *Escherichia coli*, and *Klebsialla pneumonia*. All aforementioned benefits evidenced its selection excellent as coating material. Aloe vera consists of active compounds in its profile such as vitamins, enzymes, minerals, sugars, lignin, saponins, salicylic acid, and amino acids.

### 2.2. Lemongrass Extract

Lemongrass consists of bioactive compounds which are beneficial for health. Lemongrass contains terpenoid compounds such as geranial, linalool, neral, pinene, myrcene, and terpinene. The terpene helps in the degradation of bacteria and causes toxic effects on cell membrane and cytoplasm. The antibacterial activity causes structural changes and cell lysis. Therefore, lemongrass assists in preservation and shelf-life extension. Lemongrass also contains allelochemicals which affect the insects thus also known as biopesticide. Lemongrass insecticidal properties are referred to as bioactive acyclic and cyclic terpenes. It can kill insects at the larval stage, such as cabbage looper [37], and distract insects due to the presence of caryophyllene (0.57%), germacrene D (2.24%), and caryophyllene oxide (0.58%). Lemongrass’s effective utilization stops the growth of microbes [38]. Lemongrass is useful to minimize the diseases as an edible coating in fresh whole fruit and cut fruit such as Fuji apples [39], pomegranate arils [40], fresh-cut pineapple [41], and strawberry [42].

### 2.3. Neem Extract

Neem extract has antifungal and antibacterial properties, which come from the Meliaceae family. This plant has antioxidant, microbial, and therapeutic properties. Its extract contains several bioactive compounds such as azadirachtin (the best limonoid compound [43,44], salannin, nimbidin, margolonone, gedunin, and others, which are applied as insect mite repellents. In an experiment, it was applied as an edible coating on tomatoes, where it plays a crucial role in improving texture, color, and flavor [45]. Neem extracts are famous in India for medicinal properties and are obtained from various parts of the plants and found to contain polyphenols (e.g., tannins, lignins, and flavanoids) possessing strong antioxidant [46,47], antibacterial [48,49], as well as anti-inflammatory and immunomodulatory properties [50,51]. A study showed antibacterial properties of neem extract against *Staphylococcus aureus*, *Pseudomonas aeruginosa*, *Klebsiella pneumoniae*, and *Salmonella typhi*, and a 90% inhibition rate received [52]. Phenolic compounds in neem extract profile exhibit high antioxidant properties than synthetic antioxidants [45,46]. There is a major shift towards natural extracts from synthetic chemicals due to their harmful effects on human health and the environment [53].

## 3. Use of Synthetic Chemical Pesticides

There are many advantages of pesticides, including increasing crop yield, managing vector/diseases, and killing or inhibiting hazardous pests. Despite the beneficial effects of pesticides, the adverse effects of pesticides should not be unnoticed. Both the atmosphere and human beings are seriously in danger by pesticide usage. Pesticides also disturb nature, water, and soil and cause harmful effects, leading to a dangerous impact on livestock, birds, plants, and human beings [54]. Biodiversity is often disrupted by pesticides, and continuing direct or indirect exposure to pesticides can pose serious risks to human health. Acute health conditions such as cancer, diabetes, reproductive disorders, respiratory disorders, and neurological disorders can be caused by them [55]. The abused use of harmful chemical pesticides in the agricultural environment makes it difficult to distinguish between the effects on the environment and the effects on the health of the consumers.

The lack of proper management of the fruits after harvest causes tremendous economic losses, growing poverty, hunger, and malnutrition. Globally, various postharvest technologies and synthetic chemical treatments have been used to minimize postharvest losses, but these are recorded to increase the risk to human health and the environment [56]. As an effective and cost-effective method for pest control, pesticides are deliberately considered. However, not only do they kill the target creature due to their mechanism of action, but they also damage non-target creatures, including humans. About 3 million annual cases of pesticide poisoning and up to 220,000 deaths are recorded by the World Health Organization, primarily in developing countries [57]. Additionally, particularly young and developing creatures are highly susceptible to their adverse effects due to the non-specific use of pesticides and unintentional awareness. The toxic effect of pesticide exposure was primarily determined by the quantity of pesticide as well as the persistence duration. Some pesticides are highly poisonous to humans, with just a few drops in the mouth or on the skin resulting in adverse result. Their persistent and long-term exposure to other less toxic pesticides may also cause adverse effects. In most developed countries, pesticide residues in fruits, vegetables, and food have been tracked for decades, while those in developing countries are not properly recorded [56].

### 3.1. Synthetic Chemical Pesticides: Effects on Human Health

The research found an increase in the incidence of prostate, breast, bladder, lung, colon leukemia, and multiple myeloma cancers due to constant exposure to certain chemicals and pesticides [58]. A meta-analysis has shown that exposure to pesticides can cause genetic alteration in the genes involved in pathogenesis. Environmental exposures to pesticides have raised the likelihood of changes in genes, including the GST, PON-1, MDR-1, and SNCAA [59]. The implications of different studies have established the association between pesticide exposure and diabetes. The risk of diabetes has been exacerbated by constant interaction with pesticides. A strong association between organochlorine compounds and diabetes was found, and the incidence of type 2 diabetes was similarly associated with organophosphate [60]. Elevated concentrations of dichlorodiphenyltrichloroethane (DDT) and heptachlor epoxide pollutants in human blood have been related to diabetes and related nephropathy [61]. Increasing research has linked obesity to chronic pesticides such as DDT, its DDE metabolite, and other harmful chemicals such as nitrosamines, benzoates, sulfites, sorbates, parabens, formaldehyde, BHT, and BHA [62]. A specific instance population examination in Bang Rakam demonstrated an increased risk of diabetes caused by exposure to pesticides [63]. They found that three insecticides (endosulfan, mevinphos, and Sevin) and a fungicide (benlate) were the main cause of diabetes type 2 mellitus. The preparation of imazamox-based herbicides reduced bislet cell size and triggered an increase in blood sugar and calcium [64]. Exposure to pesticides may raise the risk of lung diseases, as well as the issue of morbidity and mortality [65]. Occupational exposure among agricultural producers was associated with a higher incidence of pulmonary symptoms, including lung dysfunction, and a progressive incidence of chronic respiratory diseases [66]. Pesticides, which have been closely linked to immune disorders, have been documented to increase the risk of cancer, respiratory problems, organ ailments, system defects, nervous system abnormalities, and asthma [67]. The link between the use of pesticides and thyroid cancer has been tested in many epidemiological studies. The prevalence of thyroid cancer is high relative to the normal community throughout the Agricultural Health Study (AHS) cohort [68].

### 3.2. Synthetic Chemical Pesticides: Effects on the Environment

Pesticides as well as other chemical products have been an integral part of modern agricultural production systems. Over the past century, these chemicals have contributed to a dramatic increase in crop yields through insect and disease management [69]. The rapid expansion in the world population also emphasizes the need to maximize the supply of food. Furthermore, the widespread use of these pesticides has many negative impacts, ranging from environmental pollution to ecosystem destruction [70,71]. Environmental and agricultural land diffusion of agrochemical pollutants causes catastrophic ecosystem pollution (i.e., dust, air, soil, sediments, and water) and spoilage of human food across the globe [72]. Soil, water, grass as well as some other flora may be contaminated by pesticides. In addition to killing insects or weeds, other animals such as birds, fish, beneficial insects, and non-target plants might be contaminated by pesticides. In general, insecticides are by far the most hazardous type of pesticide, but herbicides may also pose risks to non-target species [73].

A large number of pesticides and other chemicals have been released into the environment due to agricultural and commercial growth [74]. In current agricultural practices, the widespread use of chemical fertilizers has resulted in the pollution of various environmental matrices, including air, soil, and water. Resultantly, human health and non-targeted organisms are adversely affected by polluted environmental geometries in many ways. Mostly through soil leaching, surface run-off, underground run-off, and accidental leakage, traces of these toxic compounds may enter rivers, surface water, and groundwater [75].

## 4. Postharvest Quality of Fruits and Vegetables and the Role of Plant Extracts

Fruits and vegetables, for their decent palate and nutrient richness such as polyphenolic compounds, vitamins, and organic acids, are considered to be preferred food for humans worldwide [76]. These products undergo numerous hasty fluctuations in their taste and nutrients, thus influencing consumer acceptance during postharvest storage, which ultimately results in the loss of fresh produce [76]. Senescence is a stage where anabolic processes give way to the catabolic process that is responsible for the decay of fresh produce. That leads to the undesirable modification in the postharvest quality and composite progression that is prejudiced by both endogenous and exogenous environments [77]. The significance of fresh fruits and vegetables is calculated based on mass. Due to shrinkage, a mass decline has adverse effects and massive financial losses [78]. With the perishable nature of most fruits, it is of utmost importance to seek for environment-friendly treatments to mend the quality of such products [11]. Plant extracts have recently gained a great deal of attention due to their various health benefits and possible agricultural implementations. Several previous studies have shown that the treatment of plant extracts enhances fruits and vegetables’ durability after harvest and increases shelf life. Postharvest treatments for plant extract retain higher non-enzymatic antioxidant activity, increased antioxidant capacity, hormone bio-synthesis regulation, and delayed breakdown of the cell wall [56].

### 4.1. Aloe Vera Extract: Effect on Postharvest Quality of Fruits and Vegetables

For the treatment of banana fruit, aloe vera extract delays the vicissitudes in loss of weight, titratable acidity, accumulation of soluble solids, and firmness. Moreover, spraying aloe vera gel improves the total phenolic contents and total antioxidant activities of banana fruit [79,80]. Spraying of table grapes (*Vitis vinifera*) of Shahroudi cultivar with aloe vera has an optimistic result on the postharvest quality and improves the characteristics of the weight loss, total soluble solids, berry firmness, enzymatic activity, and titratable acidity [81]. *Aloe vera* gel treatment is perceived to be an environment-friendly non-chemical substitute method for the handling of litchi fruit after harvest because it prevents browning, which would be the key restriction on the quality of litchi fruit after harvest and the selling of litchi fruit [82]. Aloe gel curtails the decay prevalence and lessens the loss of weight, rate of respiration, and ethylene production to a larger range. It also suppresses diseases and preserves the expected properties of mango fruit during postharvest storage [83]. Higher pH, soluble solids concentration, ascorbic acid, titratable acidity, total carotenoid content, total flavonoid content, and total phenolic content are preserved by treatment of papaya fruit covered with *Aloe vera* gel compared to uncoated papaya fruits that decomposed during 12 days of storing [84]. Pomegranate spraying with 100% aloe vera extract is the greatest operative for the reduction of physiological losses in terms of weight, almost 50% less reduction compared to untreated ones. *Aloe vera* extract-treated pomegranate fruits show reduced total soluble solids to the acid ratio (32.17%) and pointedly maximum juice contents (47.17%), ascorbic acid (12.82 mg/100 g), and anthocyanin (13.98 mg/100 g) of the fruits as well as the maximum organoleptic rating [85]. Sapodilla fruit coated with aloe vera keeps tall firmness and titratable acidity levels compared to uncoated fruits [79]. After harvesting, aloe vera on the blueberry surface offers an extra shield to minimize postharvest microbial spoilage as well as to decrease the loss of water, the two key factors of postharvest blueberry loss in quality. A schematic diagram of aloe vera extract utilization is shown in Figure 1.

### 4.2. Lemongrass Extract: Effect on Postharvest Quality of Fruits and Vegetables

It is suggested that lemongrass essential oil sustains the quality traits of fruits throughout postharvest handling. There is a positive impact of lemongrass essential oil [86] on the postharvest quality characteristics of apple cultivars ‘Granny Smith’ and ‘Pink Lady’. The appraised phenolic contents and antioxidant activity of fruits convey health benefits to consumers [26]. Lemongrass essential oil spraying on blackberries in combination with micro-fibrillated cellulose at concentrations of 0.2, 0.4, and 0.6% are effective in protecting blackberry fruits’ freshness and dropping the color reversal while in stores [87]. Applications of lemongrass essential oil significantly preserved the physical quality parameters of the cucumber at an ambient temperature of 28 ± 2 °C during the postharvest storage [88]. Guava fruit treated with lemongrass essential oil preserves and maintains its physicochemical characteristics during postharvest handling [89].

### 4.3. Neem Extract: Effect on Postharvest Quality of Fruits and Vegetables

The treatment effect of neem leaf extract (20% or 40%) on the postharvest quality of the banana has a significant effect on the physicochemical parameters of the banana fruit [90]. Neem-extract-treated mangoes show expressively the lowest physiological losses in weight. Such treatment suggestively improves and maintains the better physic-chemical quality of Amrapali mango fruit [91]. The neem leaves extract treatment at 5% concentration imparts the best positive effect on the external quality of papaya fruit. Papaya Formosa fruits treated with neem extract keep their good quality during postharvest handling and storage [92]. This postharvest procedure of applying neem extract to the pitaya fruit has a massive impact on the freshness of pitaya in terms of commercial exploitation with the best postharvest quality and good taste with high antioxidants [93].

## 5. Plant Extracts’ Influence on Postharvest Diseases and Shelf-Life Extension

The food manufacturing sector is always in front of the challenge of food storage problems and intensifying the life of fresh fruits and vegetables after harvest. Foodborne pathogens are one of the chief agents and are responsible for the spoilage of food and the worsening of consumer health [94]. Table 1 shows the influence of plant extracts effect on the postharvest quality of different fruits. Fruits and vegetables are living creatures, and during breathing, they continuously use oxygen and produce carbon dioxide. The metabolism of substrates such as proteins, fats, and organic acids also with carbohydrates takes place in the respiration phase. When substrates and carbohydrates are metabolized, then it becomes challenging to replenish them since the vegetable or fruit is removed from the plant [95].

There is a rising awareness about the use of natural and safe plant-based preservatives for the prolongation of fruits and vegetables to substitute harmful chemicals to combat the challenges of preserving food with less or no harm to consumer health [114]. Fruits and vegetables are consumed processed or fresh according to their various health benefits [95]. The growing claim for pesticide-free food has pushed the research towards the era of plant-based extracts as a resolute substitute for hazardous pesticides [18]. Natural polymers have various advantages compared to synthetic polymers due to their biocompatibility, biodegradability, and compliance with chemical and biochemical modifications [115]. A schematic diagram of plant extract utilization against postharvest decay is shown in Figure 2.

### 5.1. Aloe Vera Extract: Effect on Postharvest Diseases and Shelf-Life

*Aloe* gel spraying can be used to regulate and maintain the inhibition of banana fruit metabolism, leading to anthracnose disease resistance. An alternative technique to synthetic pesticides to combat anthracnose disease and preserve the postharvest quality of banana fruit can be maintained by *Aloe* gel spraying [79]. Chitosan-based aloe vera extract edible spraying lengthens the shelf-life of Blueberries up to 5 days, indicating that the chitosan and aloe vera liquid combination potentiates the shelf-life extension of fruits. *Aloe* gel positively influences storage life and table grapes’ postharvest quality. *Aloe vera* extract application has proved to be an eco-friendly alternative technique to synthetic chemical treatment for the postharvest quality maintenance of litchi fruit. *Aloe vera* gel-coated fruits and vegetables show reduced postharvest browning, the leading limitation for prolonged storage life and presentation of litchi [82]. Chitosan-based aloe vera gel applications prolong the postharvest quality and the storage life of mango fruit [83]. Papaya fruits coated with aloe vera extract exhibit a significant delay in ripening, stifled the growth of fungus, and preserved the postharvest papaya fruits’ quality in storage for up to 15 days [84]. The spraying of *Aloe vera* extract considerably expands the storage life along with the improved retentive fruit quality features under normal storage conditions [82]. In addition, the aloe vera antifungal compound provides an innovative way of enhancing the protection and shelf-life of blueberries without the need for synthetic substances in edible films or spraying. *Aloe vera* fractions may be an enticing healthier solution to fungi that invade fruit and vegetables, avoid unnecessary use of additives, and thus help to prevent health and environmental issues from occurring.

### 5.2. Lemongrass Extract: Effect on Postharvest Diseases and Shelf-Life

Lemongrass essential oil comprises numerous antifungal properties and is an effective and workable choice in postharvest treatment, as it interposes the growth of microorganisms. Lemongrass oil at 1.0% concentration was found to be a very effective postharvest treatment against anthracnose without phytotoxicity effect on banana fruits [99]. Lemongrass essential oil treatment inhibits the crown rot pathogens of banana fruit. Lemongrass oil shows an antifungal effect in contradiction to *C. musae*, *Fusarium incarnatum*, and *F. verticillioides* at 475, 350, and 600 μL/L, respectively [100]. The potential of lemongrass essential oil for enhancing the shelf life can be utilized on blackberries. The application of essential oil reduces the deterioration of fruit and prolongs the storage life for longer consumption periods [65]. Guava fruit treated with lemongrass extract shows a great effect on the management of *Colletotrichum gloeosporioides.* The inhibition of mycelial growth and sporulation of the pathogen can be observed at different concentrations of the plant extracts. However, complete inhibition was achieved at an 8% concentration [101]. Applying lemongrass essential oil vapors at different concentrations has significant potential to control the *Penicillium* decay of orange [104]. Lemongrass extract has antifungal activity against *A. niger* and *C. musae* [116].

### 5.3. Neem Extract: Effect on Postharvest Diseases and Shelf-Life

Neem extract facilitates the storage period of banana fruit. Banana fruit applied with neem extract show good storage quality. The application of 40% neem extract enhances the shelf life of banana fruit for a considerably longer period [90]. Neem extract can be used to successfully manage postharvest losses due to postharvest diseases of fresh produce in other tropical and subtropical fruits. Application of neem extract inhibits the mycelial growth of pathogens *Neofusicoccum parvum*, *Lasiodiplodia theobromae*, *Aspergillus flavus*, *Botrytis cinerea*, and *A. niger* in mangoes and longer the shelf life of fruit. Postharvest treatments of mango fruit with neem extract meaningfully affect the microbial parameters of mango fruit. Treatment of neem extract encourages storage stability and enhances the period of storage [106].

### 5.4. Other Plant Extracts and Their Effect on Postharvest Quality and Shelf Life

The efficacy of the edible spraying of the soy and wheat gluten-based plant protein extract preserves the consistency and shelf life of the strawberry postharvest quality. This treatment maintains ascorbic acid, which is related to the contents of vitamin C of the strawberry fruit, the chief component to determine the activity of the antioxidants of strawberry [78]. Applying and spraying seed/leaf extracts of moringa leaf and 1% carboxymethylcellulose on avocado fruit cultivars ‘Hass’ and ‘Gem’ has a great influence on postharvest quality and shelf life. Throughout the storage period, treatment decreases ethylene output, respiration rate, and greater firmness compared to un-coated fruit. This application suggests a decreased loss of moisture, the occurrence of disease, and the severity of the coated fruit. Spraying treatments extend the shelf-life and preserve the consistency of the two avocado cultivars after harvest. In addition, moringa ethanol extract had significant antimicrobial activity and was far more effective as compared to methanolic extracts in hindering foodborne pathogens. Organic postharvest treatment for ‘Hass’ and ‘Gem’ avocado could be a 1% carboxymethylcellulose edible spraying comprising moringa leaf/seed extracts [117]. The *Penicillium digitatum* mycelial growth proliferation and in vitro sporulation and green mold on oranges are reduced by volatile substances of brassica sachets and extracts. With canola sachets, green mold control is achieved, and its conjunction with thermotherapy does not seem to improve the condition. Canola sachets have a great capacity as potential prevention for green mold and should therefore be tested under conditions of bacterial invasion as well as other postharvest disorders as well [118]. Dukung anak extrude crude or turmeric extracts can be used as a bio fungicide to control postharvest anthracnose in dragon fruit at a total concentration of 10.0 g. Furthermore, fruits treated with ginger and turmeric can cure the original color, aroma, and skin of the fruits and vegetables [119]. The application of edible spraying and plant extracts for pre-storage can be effective in preventing maturation and ensuring the integrity of guava. The mixture of gum Arabic and garlic extract suppressed weight loss, skin browning, disease progression, and prolonged shelf life of guava fruits under environmental conditions. Overall, the combined treatment of Arabic gum and garlic extract still preserves the physiological, chemical, and sensory consistency of the guava fruit. Such treatment of garlic extract and gum Arabic may also be considered an efficient treatment for the extended storage life and the consistency of guava fruit preservation [96]. A schematic diagram of the beneficial effects of spraying on fruit quality is shown in Figure 3.

Many notable findings strongly suggest the noteworthy function of plant oligosaccharides in the sustainability of fruit after harvest, which places importance on oligosaccharide care, retaining the consistency of fruit, and preserving the life of storage by maintaining various physicochemical properties. Plant oligosaccharides have enhanced the function of antioxidants, the activity of enzymes relevant to disease defense, and control of the maturation of gene expression [56]. The combination of treatments with chitosan, oligosaccharide, and salicylic acid could substantially delay the maturation of apricots. In addition, during storage, the application of chitosan, oligosaccharide, and salicylic acid retained postharvest consistency characteristics of the ‘*Xiaobai*’ apricot, involving latency in the increasing rate of decay, loss of firmness, and a decline in total soluble solids and titratable acidity content, and color shift [120]. The taste, odor, and color of some plant extracts used for the preservation of food might change their organoleptic properties [121]. However, in some cases, it is vice versa, like extracts improve the quality of the food [122]. One can also tackle this limiting factor of plant extracts by removing the coloring agents from extracts and using the specified dose of plant extracts [123].

## 6. Conclusions

The continuous growth of the population has driven the increased demand for fresh fruits and vegetables; that is why plant-based extract could be a decent alternative source to the chemical synthetic pesticides for preserving the postharvest quality of fresh fruits and vegetables and for the enhancement of their shelf life. Chemical pesticides have been used for a long time in the agriculture sector for the destruction of pests exert hazardous consequences on both environment and human health. The research explored the health hazards of these chemicals. Natural alternatives to chemicals are safer and perform the same role as pest control and postharvest quality maintenance. *Aloe vera*, lemongrass, and neem extract are economical sources of diverse, high-added-value materials that exhibit the potential to maintain the postharvest quality of fruits and vegetables and good natural alternatives to hazardous chemicals as well. These promising natural extracts can be an excellent source alternative to chemical pesticides and preservatives to protect fresh fruits and vegetables from spoilage.

## Figures and Tables

**Figure 1 molecules-27-05152-f001:**
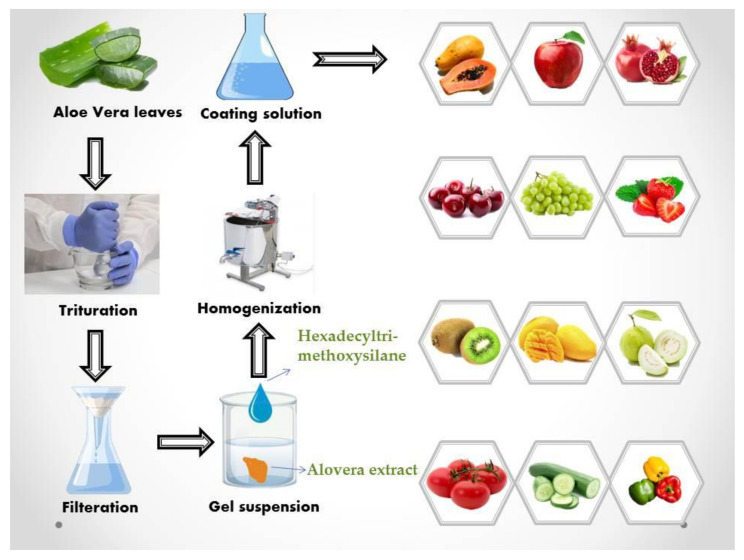
*Aloe vera* extract as a spraying material.

**Figure 2 molecules-27-05152-f002:**
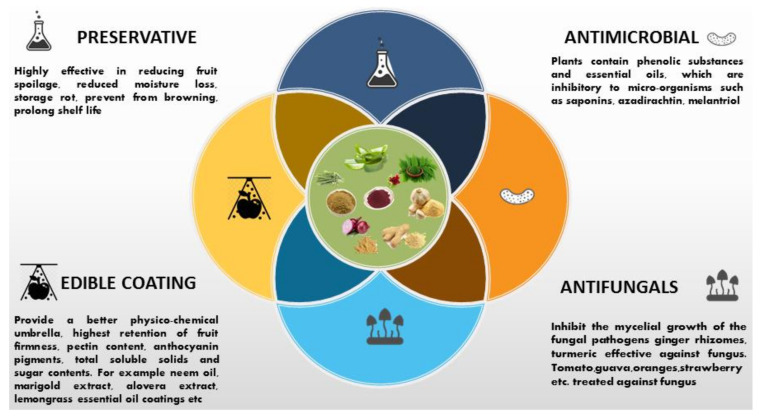
Plant extracts utilization against postharvest problems.

**Figure 3 molecules-27-05152-f003:**
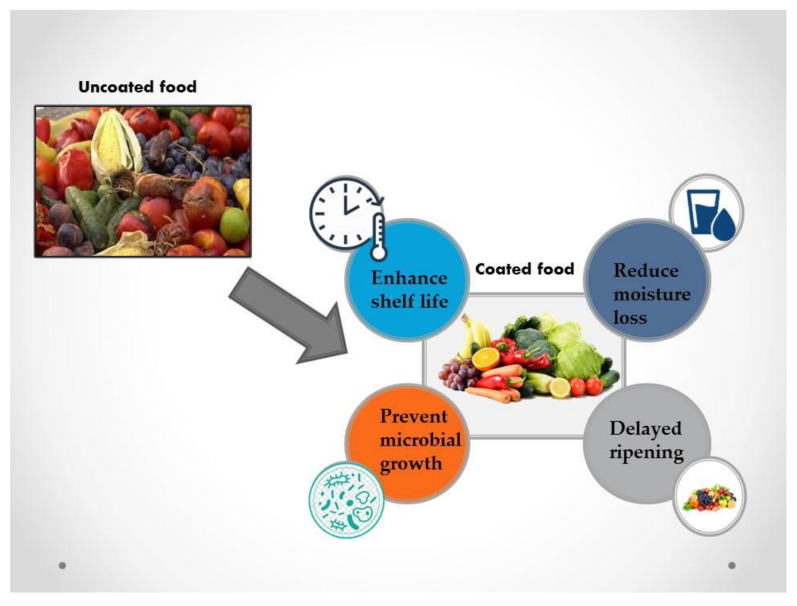
Beneficial effects of spraying on fruits and vegetables.

**Table 1 molecules-27-05152-t001:** Plant extracts’ effect on postharvest quality.

Plant Extract	Fruits	Treatments	Results	Reference
*Aloe vera* (*A. barbadensis* *Miller)*	Sapodilla (*Manilkara zapota* L.)	*Fagonia indica* 1% + aloe vera gel 100%	Radical scavenging activity improved Maintained total phenolic and flavonoids Higher ascorbic acid maintained No negative effect was found on sensory attributes Shelf life improved for 12 days	[79]
	Guava (*Psidium guajava*)	10% garlic extract + 100% aloe vera gel	Higher total flavonoid Enhance shelf life	[96]
	Grapes (*Vitis*)	20% aloe vera	Reduce moisture loss Provide a good gloss appearance Reduce fungal infection Enhance shelf life	[97]
	Banana (*Musa acuminate*)	Aloe vera gel 0.05% + garlic oil 0.1%	No change in weight loss Delay firmness, soluble solids, and titratable acidity Enhanced total phenolic contents and total antioxidant	[85]
	Litchi (*Litchi chinensis*)	Aloe vera gel 50%	Maintained postharvest quality Reduced browning index and weight loss Higher ascorbic acid content Increase total phenolic concentration	[82]
	Mango (*M. indica*)	1% chitosan + 1% aloe vera gel	Prevent from decay Reduced weight loss and ethylene production Maintained the natural properties of mango	[83]
	Papaya (*Carica papaya*)	50% aloe vera gel	Enhance phenolic content (41.75 mg/100 g) Shelf life enhances up to 9 days	[84]
	Blueberry (*Vaccinium corymbosum*)	Chitosan 0.5% + aloe vera liquid fraction 0.5%	Microbiological growth reduced 50% water loss minimized Mold contamination appears with a gap of 9 days Shelf life improved for 5 days	[34]
	Object with Gram-Positive bacteria	30 μL *Aloe vera* leaf extract	Inhibition of *B. subtitis* (15 mm) Inhibition of *B. cereus* (13 mm) Inhibition of *B. megaterium* (14.5 mm) Inhibition of *Streptococcus pyogenes* (13 mm) Inhibition of *Staphylococcus aureus* (14 mm)	[98]
	Object with Gram-Negative bacteria	30 μL *Aloe vera* leaf extract	Inhibition of Echerichia coli and Agrobacterium tumefacins (18 mm)	
	Object with Gram-Positive bacteria	30 μL *Aloe vera* root extract	Inhibition of Bacillus subtitis, B. megaterium, and Enterococcus faecalis (16 mm) Inhibition of *B. cereus* (13.5 mm)	
	Object with Gram-Negative bacteria	30 μL *Aloe vera* root extract	Inhibition of *Agrobacterium tumefacins* (17.5 mm) Inhibition of *E. coli* (16 mm)	
	Object with fungus	30 μL *Aloe vera* leaf extract	Inhibition of *Fusarium oxysporum* (18.5 mm) Inhibition of *Aspergillus niger* (18 mm)	
	Object with fungus	30 μL *Aloe vera* root extract	Minimum but good results for inhibition of *Fusarium oxysporum* and *Aspergillus niger*	
Lemongrass oil (*Cymbopogon citratus*)	Banana (*Musa paradisiaca* Linn.)	1.0% lemongrass oil + 2.0% neem oil	Reduce fungal growth Inhibit the *Colletotrichum musae* which cause Anthracnose in Banana fruit	[99]
	Banana (*M. paradisiaca* Linn.)	41.29% lemongrass oil + 32.15% geraniol	Effective against *C. musae*, *Fusarium incarnatum,* and *F. verticillioides* Inhibited mycelial growth and conidia germination	[100]
	Guavas (*Psidium guajava* L.)	Chitosan + 1.5% essential oil	Inhibition of anthracnose lesions Enhance the stability of the guavas fruit for up to 12 days	[101]
	Apple (*Malus domestica*)	Thermal fogging −0.5 °C + lemongrass essential oil 1.5% + control atmosphere	Lower titratable acidity and total soluble solid content Higher total phenolic content and radical scavenging activity	[102]
	Blackberry (*Rubus fruticosus*)	Lemongrass essential oil 1000 ppm + micro fibrillated cellulose 0.4%	No changes in total soluble solids	[103]
	Guava (*Psidium guajava*)	0.4 kGy γ irradiation + lemongrass oil 2%	Reduced decay and water loss percentages Controlled total soluble slid, titratable acidity, and vitamin C decrements Increased fruit softness	[89]
	Orange (*Citrus sinensis* L. osbeck)	Red thyme oil 6.7 μL/L + essential oil	*Penicillium* decay Inhibit the production of spore Extend shelf life up to 12 days	[104]
Neem (*Azadirachta indica*) (LD50 ≥ 5000 mg/kg)	Banana (*M. paradisiaca Linn*.)	40% neem leaf extract	Minimal color change No disease incidence Lower reduction in titratable acidity Enhance longer shelf life 8.33 days	[90]
	Mango (*Mangifera indica*)	40% of neem leaf extract + 40% banana pulp	Slower changes in color (score 2.93) Firmness (score 2.77) Less disease severity (score 3.57) Disease incidence (60.00%) Lower loss in weight (35.17%) Longer shelf life (10.25 Days)	[105]
	Mango (*M. indica*)	Crude water extract of neem 30%	Control mango fruit rot incidence	[106]
	Tomatoes (*Lycopersicon esculentum Mill*.)	25% neem leaf extract	Enhance shelf life Effectively reduce weight loss by 55–60% Change in titratable acidity 20–67% Total soluble solubility 2.8–6.9	[107]
	Papaya (*C. papaya*)	5 and 10% neem leaf extracts	5% had the best effects on external quality Inhibit the growth of phytopathogenic fungi Enhance shelf life for 12 days	[92]
Moringa (*Moringa oleifera*)	Avocado (*Persea Americana*)	1% carboxyl methylcellulose + 2% *moringa* leaf extract	Lower mass loss, ethylene production, and respiration rate Inhibition of gloeosporioides and *A. alternate* Higher antimicrobial activity Prolongs the shelf-life	[26]
Green tea (*Camellia sinensis*)	Potato (*Solanum tuberosum*)	50 mL/L green tea extract	Controlled browning in fresh-cut potatoes Prolong shelf life for 14 days	[108]
Clove (*Syzygium aromaticum*)	Potato (*S. tuberosum*)	1–25 g GAE/L clove extract	Higher total phenolic content Improve antioxidant activity >50% inhibition of potato polyphenol oxidase	[108]
Pomegranate peel (*Punica granatum*)	Apricot (*Prunus armeniaca*)	Chitosan + 1% pomegranate peel extract	Reduced decay percentage and weight loss Effectively retained DPPH radical scavenging activity Improve firmness Enhance shelf life	[109]
Pomegranate peel extract (*Punica granatum*)	Sweet cherry fruit (*Prunus avium*)	CaSO4 1% + Pomegranate peel extract 400 ppm	Preserved fruit colour during storage Enhance shelf life	[110]
*Prosopis juliflora* water-soluble leaf ethanolic	Strawberry (*Fragaria x ananassa*)	1% chitosan + *Prosopis juliflora* leaf extract	Maintenance of firmness Improve total soluble solids Inhibit the microbial load Lower percent weight loss Increase total antioxidant levels	[111]
Blackberry (*Morus nigra* L.)	Cherry tomato (*Solanum**Lycopersicum,* L.)	Blackberry (*Morus nigra L*.) anthocyanin rich-extract 30%	Maintain constant fruit weight and firmness Increase shelf life	[112]
Thyme (*Thymus vulgaris*)	Avocado (*Persea Americana*)	Applied 2000 ppm	Decrease lesion expansion Increase firmness Enhance shelf life	[113]
Guava Leaf (*Myrtaceae*) and Lemon extract (*C. limon*)	banana cv. Sabri (*Musa sapientum* L.)	20% Guava leaf and 15% Lemon extract	Enhance shelf life for long-term storage	[90]

## Data Availability

Not applicable.

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
