# Peer review of "Natural Plant Extracts: An Update about Novel Spraying as an Alternative of Chemical Pesticides to Extend the Postharvest Shelf Life of Fruits and Vegetables"

_molecules, 2022, doi:10.3390/molecules27165152_

Round 1

Reviewer 1 Report

Manuscript provide review about applications of plant extracts for coating materials for fruits and vegetables.

Title and abstract could be more concise.

Plant extracts in the introduction of chapters could be more characterized.

Use of abbreviations for plant extracts (e.g. „AV“ for aloe vera/LEO/EO for lemongrass oil extract) or some others (e.g. „CMC“ in page 11 or „TSS“ and „TA“ in page 12 line 402) are unsystematic and unclear that  impairs good readability.

English language and syntax should be deeply revised.

Notice to some errors and mistyping:

e.g. Fig. 1 „alovera leaves“

Table 1: units “dm3 kg-1“

Concentration units: 10.0 g in page11 line 379

The end of sentence in page 7 line 253 is miising

Taxonomy should be typed by italica e.g. in page 11 line 335-336, 340, 352-353

Revise and complete References carefully and include novel sources.

The images and summary tables are captivating, however there are some suggestions for improving clarity and comprehensibility:

-       arrangement and logic (e.g. the position of the rings in Fig. 4 should be placed with the image of fruit with coating)

-       some elements could be confusing (e.g. in Fig. 1: some orange object in the aloe vera extract, or some spiral above the surface of the liquid; unknown abbreviation „HDTMS“)

-      consider focus and arrangement of Fig. 2 (e.g. characterize preservative effects of edible coatings)

Author Response

Dear Reviewer,

Thank you very much for the kind review of our manuscript. We appreciate the opportunity to make changes. I assure You that all comments indicated by You were carefully considered and included in the revised version of the manuscript. The detailed answers to your comments and suggestions are uploaded in a separate file.

We appreciate the opportunity to make changes, we hope that after revising, the manuscript in the new form will be accepted for publication in Molecules.

Suggestions for Authors

Manuscript provide review about applications of plant extracts for coating materials for fruits and vegetables.

Title and abstract could be more concise.

Response: Title has been shortened as per our understanding.  

Plant extracts in the introduction of chapters could be more characterized.

Response: We think plant extracts already characterized as much according to available literature.

Use of abbreviations for plant extracts (e.g. “AV” for aloe vera /LEO/EO for lemongrass essential oil) or some others (e.g. “CMC” in page 11 or “TSS” and “TA” in page 12 line 402) are unsystematic and unclear that  impairs good readability.

Response: Replaced with complete words throughout the manuscript.

English language and syntax should be deeply revised.

Response: English language has been modified and revised.

Notice to some errors and mistyping:

e.g. Fig. 1 „alovera leaves“

Response: Modified.

Table 1: units “dm3 kg-1“

Response: Modified.

Concentration units: 10.0 g in page11 line 379.

Response: Modified.

The end of sentence in page 7 line 253 is miising

Response: Modified.

Taxonomy should be typed by italica e.g. in page 11 line 335-336, 340, 352-353

Response: Modified.

Revise and complete References carefully and include novel sources.

Response: Modified.

The images and summary tables are captivating, however there are some suggestions for improving clarity and comprehensibility:

-  arrangement and logic (e.g. the position of the rings in Fig. 4 should be placed with the image of fruit with coating)

Response: Modified.

-  some elements could be confusing (e.g. in Fig. 1: some orange object in the aloe vera extract, or some spiral above the surface of the liquid; unknown abbreviation “HDTMS“)

Response: Modified.

-  consider focus and arrangement of Fig. 2 (e.g. characterize preservative effects of edible coatings)

Response: Diagram is arranged and already mentioned about the preservative effects of edible coating in (preservative portion).

Reviewer 2 Report

Dear reviewers ,

This is an interesting review that focused on natural plant extract to extend the postharvest shelf-life  of fruits and vegetables. There are some points that needs to be attended.

Title. It is so large, consider shortening.

Abstract. Must be less ambiguous and more precise about the topics that will be specifically addressed.

Introduction. There is a lack of references here, most indeep research to introduce the work must be done.

Results. The  topics 2 about synthetic pesticides can be summarized in the introduction section, including only  1 or 2 paragraphs giving a novel and hard data... e.g. people contaminated with pesticide  worldwide, number of banned pesticides each year, etc.... Otherwise, it would be confusing, given the current title I would expect to start with natural product properties, etc. Pesticide issues are widely known. 

The table 1 and 2 you should mention AT LEAST 15 examples on each onethe , from latest papers (5 last years). And them make a critical discussion of these tables.}

Why did you select those extracts? randomly?

Line 234 and inside table put the long name of Aloe vera instead AV. For each extract,  you must mention which are the main actives... Have in mind that polymeric materials are used as a coating but natural extract rich in low molecular weight compounds does not form a coating barrier around the fruit, just used as regular spraying like pesticides.Please correct this along manuscript.

You must mention the LD50 found for each component. Are they considered as GRASS, please check it and provide information about them. I wonder if Neem oils or extracts are GRASS? Did you check this?. In a negative case, it will be considered a regular pesticide.

It looks like the postharvest discussion is fractionated into small pieces to discuss something; sometimes, it is repetitive. I strongly suggest merging some parts and having a proper discussion. Also, suggest adding more extracts used as postharvest treatments (at least 2 or 3)  . Ones with more commercial potential or uses to enrich this discussion. The information about those selected extracts is scarce and narrow. 

Conclusions must be rewritten.

Author Response

Dear Reviewer,

Thank you very much for the kind review of our manuscript. We appreciate the opportunity to make changes. I assure You that all comments indicated by You were carefully considered and included in the revised version of the manuscript. The detailed answers to your comments and suggestions are uploaded in a separate file.

We appreciate the opportunity to make changes, we hope that after revising, the manuscript in the new form will be accepted for publication in Molecules.

Title. It is so large, consider shortening.

Response: Shortened

Abstract. Must be less ambiguous and more precise about the topics that will be specifically addressed.

Response: Modified as per our understanding.

Introduction. There is a lack of references here, most in deep research to introduce the work must be done.

Response: More references has been added. According to the next below comment, we have added new data from topic 2.

Results. The topics 2 about synthetic pesticides can be summarized in the introduction section, including only 1 or 2 paragraphs giving a novel and hard data... e.g. people contaminated with pesticide worldwide, number of banned pesticides each year, etc.... Otherwise, it would be confusing, given the current title I would expect to start with natural product properties, etc. Pesticide issues are widely known.

Response: We have changed it according to your kind suggestions.

The table 1 and 2 you should mention AT LEAST 15 examples on each one the , from latest papers (5 last years). And them make a critical discussion of these tables.}

Response: Dear reviewer, we have worked very hard to compile the Table and we think there is no need to split the Tables into 3 Tables. Because there is no difference in available literature data. Most of the studies having the same factors and most of the added data from different recent research article in this Table.

Why did you select those extracts? randomly?

Response: We selected the most important, valuable, easy to use and having the medicinal value plant extracts. Now, adjusted in a sequence.

Line 234 and inside table put the long name of Aloe vera instead AV. For each extract,  you must mention which are the main actives... Have in mind that polymeric materials are used as a coating but natural extract rich in low molecular weight compounds does not form a coating barrier around the fruit, just used as regular spraying like pesticides. Please correct this along manuscript.

Response: Modified and repharased.

You must mention the LD50 found for each component. Are they considered as GRASS, please check it and provide information about them. I wonder if Neem oils or extracts are GRASS? Did you check this?. In a negative case, it will be considered a regular pesticide.

Response: Yes, we have checked it, the neem extract is a natural pesticide and practically non-toxic to birds, mammals, bees, and plants.

It looks like the postharvest discussion is fractionated into small pieces to discuss something; sometimes, it is repetitive. I strongly suggest merging some parts and having a proper discussion. Also, suggest adding more extracts used as postharvest treatments (at least 2 or 3)  . Ones with more commercial potential or uses to enrich this discussion. The information about those selected extracts is scarce and narrow.

Response: Data emerged and added more discussion.

Conclusions must be rewritten.

Response: Modified.

Reviewer 3 Report

Dear respected authors,

 I carefully checked the manuscript content and it was interesting and suitable for publication. There are some comments that should be considered before the publication of this review. Please submit an answer sheet while you re-submit the revised version of this paper. Note that all answers should be clear and deal with all comments posted on the manuscript.

·       Introduction section has only ten references. Please use more references for paragraphs that remain uncited. Citing valuable references helps academic readers to get more information about discussed materials there.

·       Please add DOI or PMID number for all cited references.

·       Please check the references for retracted papers. Overall, retracted papers should not be cited in submitted papers. You can use Zotero software for this case. Citing retracted papers will bring papers further complications.

·       Please insert literature search strategies and all keywords that were used to gather relevant data. This case helps academic readers to obtain critical data from your paper.

·       Lines 39-42: please add a reference

·       Lines 48-51: please add a reference

·       Lines 57-60: please add a reference

·       Lines 61-67: Please update the reference.

·       Lines 69-74: please add a reference

·       In the introduction, please add the global statistics on available plant-based extracts for the preservation of fruits and vegetables.

·       Please add a paragraph about health complications of utilized pesticides for the preservation of stored fruits and vegetables

·       After the introduction section, please add a literature search strategy and applied keywords for gathering data. I highly recommend you use VOSviewer software for drawing a scientific map of current findings on natural methods for the preservation of fruits.

·       Lines 115-119: please add a reference

·       Line 123: after reference 17: please go to the new line and divide the text into smaller paragraphs.

·       Lines 147-153: please add reference!!!

·       In section 2: please also discuss the clinical complications of pesticides and relevant chemicals used in the preservation of fruits and vegetables for the human gastrointestinal system.

·       Association between exposure to pesticides and the prevalence of diabetes is not still well-documented. Please find some meta-analyses or systematic reviews supporting the mentioned claim in lines 167-170.

·       DDT is one of the most dangerous chemical agents worldwide. Presently, the application of this chemical agent is banned in many geographic areas. Please exemplify other types of modern hazardous agents.

·       Lines 174-175: which pesticide? Please clearly mention which pesticides increase the prevalence of diabetes.

·       Please also describe the occurrence of which diabetes subtype is associated with exposure to pesticides.

·       Lines 176-185: Please add a table to the paper and summarize the current findings on the health complication of chemicals for the human body. Please mention several molecular signaling pathways in the human body that are susceptible to pesticides.

·       Please mention how pesticides dysregulate gene expression profile and level of transcription factors in the human body. There are several studies in this line that have not been discussed in the paper. For instance, you can see the following references:

o   Int J Environ Health Res: 2021 Nov;31(7):805-822. DOI: 10.1080/09603123.2019.1690132

o   https://doi.org/10.1016/B978-0-323-85160-2.00005-6

o   https://doi.org/10.1016/j.etap.2018.08.018

·       Please add a table to the text to compare the effectiveness of chemical agents and plant extracts for the preservation of stored fruits and vegetables.

·       Please also discuss the possible effects of the combination of safe nanoparticles with plant extracts to preserve fruits and vegetables.

·       According to your searches, essential oils are more effective for the preservation of fruits or raw extracts? Preparing plant extracts from contaminated sources with mycotoxins and other biological toxins might increase further concerns about the safety of these products. Please add a short explanation about this case to the paper.

·       Plant extracts have some special secondary products that can be easily oxidized under environmental temperature. How this disadvantage of plant-based preservative agents should be limited in storage sites?

·       Due to their natural edible properties, plant extracts can be used in fruit and vegetable preservation by dipping, fumigating, spraying, or by combining a composite coating with a carrier such as cling paper, and the effect is significant. Please discuss these methods regarding the application of plant extracts for preserving fruits and vegetables. Please also discuss which method is more effective compared to others for the protection of stored fruits and vegetables.

·       Please also add a section to discuss the limitation of plant extracts for the protection of fruits. The combinatorial methods for preserving fruits and vegetables from post-harvest pests and diseases have not been discussed in this paper. please add some lines about the possible combination of chemical and plant extracts for preserving stored fruits and vegetables.

·       One of the most important complications of plant extracts for preserving stored fruits and vegetables is their effects on the organoleptic properties of foods. A majority of plant extracts have chemical metabolites that can interact with biological components of fruits and vegetable outer peel and might reduce the quality of these products. How this limiting factor should be eliminated in the fruit industry when plant extracts will be used as protective agents? Please briefly discuss this case at the end of the review. The following reference might support the respected authors to discuss this issue:

o   https://www.frontiersin.org/articles/10.3389/fmicb.2021.753518/full

·       Although plant extracts are safe and edible products (in 98% of all cases), however, there is an import issue associated with these natural products that influenced their large-scale application in fruits and vegetable industry as protective agents. This issue is the stability of plant extracts. In comparison to presently recruited protectant chemicals, plant extracts are less stable and after administration to stored sites they might loss a significant proportion of their biological activity. Please kindly consider some references in the literature to discuss this case in your review.

·       Please check the number and caption of inserted figures. The last two figures data can be combined.

Overall, this paper provides a novel perspective on the application of plant extracts to preserve fruits and vegetables. It is well-designed and has a good structure. However, considering the above-mentioned comments before publication will boost the quality of the paper, and updating its content increase its impact on scientific readers of this journal. 

Author Response

Dear Reviewer,

Thank you very much for the kind review of our manuscript. We appreciate the opportunity to make changes. I assure You that all comments indicated by You were carefully considered and included in the revised version of the manuscript. The detailed answers to your comments and suggestions are uploaded in a separate file.

We appreciate the opportunity to make changes, we hope that after revising, the manuscript in the new form will be accepted for publication in Molecules.

Comments and Suggestions for Authors

Dear respected authors,

 I carefully checked the manuscript content, and it was interesting and suitable for publication. There are some comments that should be considered before the publication of this review. Please submit an answer sheet while you re-submit the revised version of this paper. Note that all answers should be clear and deal with all comments posted on the manuscript.

  • Introduction section has only ten references. Please use more references for paragraphs that remain uncited. Citing valuable references helps academic readers to get more information about discussed materials there.

Response: Same kind of opinion from Reviewer 1 and 2, which has been updated and taken into the consideration as per the directions of all the reviewers. More studies added with references.

  • Please add DOI or PMID number for all cited references.

Response: We have used Mandalay software as reference management software, and it will be automatically added at the proof stage. DOI has been added in the revised manuscript.

  • Please check the references for retracted papers. Overall, retracted papers should not be cited in submitted papers. You can use Zotero software for this case. Citing retracted papers will bring papers further complications.

Response: We have used 123 references in this paper and if you know any retracted paper then please let us know, we shall replace it. Last time we have published a paper and then the proof manager has suggested us to replace the paper and then we have replaced that retracted paper.

  • Please insert literature search strategies and all keywords that were used to gather relevant data. This case helps academic readers to obtain critical data from your paper.

Response: All the literature was collected from different sources through internet. The different key words related to our paper were used to search the relevant material.

  • Lines 39-42: please add a reference

Response: Reference is added

  • Lines 48-51: please add a reference

Response: Modified.

  • Lines 57-60: please add a reference

Response: Modified.

  • Lines 61-67: Please update the reference.

Response: Modified.

  • Lines 69-74: please add a reference

Response: Modified.

  • In the introduction, please add the global statistics on available plant-based extracts for the preservation of fruits and vegetables.

Response: Modified as per your direction (Plant Extracts use in the world market has a significant scope. This market is growing very fast, and the growth projected to reach USD 55.3 billion by 2026) (Reference 20). The separate data for the use in preservation is not available yet,  it will be great if you provide us the link and then we shall update it at the next revision stage.  

  • Please add a paragraph about health complications of utilized pesticides for the preservation of stored fruits and vegetables

Response: Data already available in 3.1 section of the manuscript.

  • After the introduction section, please add a literature search strategy and applied keywords for gathering data. I highly recommend you use VOSviewer software for drawing a scientific map of current findings on natural methods for the preservation of fruits.

Response: It is a very nice suggestion. But we are very sorry to say that we have not learned this software (VOSviewer) for this manuscript. We have 10 days to revise this paper and we worked very hard to complete it before the deadline. We will try to learn and use it in next manuscript in sha Allah. Thanks again for this comment.

  • Lines 115-119: please add a reference

Response: Modified.

  • Line 123: after reference 17: please go to the new line and divide the text into smaller paragraphs.

Response: Modified.

  • Lines 147-153: please add reference!!!

Response: Modified.

  • In section 2: please also discuss the clinical complications of pesticides and relevant chemicals used in the preservation of fruits and vegetables for the human gastrointestinal system.
  • Association between exposure to pesticides and the prevalence of diabetes is not still well-documented. Please find some meta-analyses or systematic reviews supporting the mentioned claim in lines 167-170.

Lines 176-185: Please add a table to the paper and summarize the current findings on the health complication of chemicals for the human body. Please mention several molecular signaling pathways in the human body that are susceptible to pesticides.

Response: It is for your kind information that our paper is totally related to Agriculture/ Food Sciences point of view. It is not related to human health etc. But data related human health already available in 3.1 section of the manuscript.

  • DDT is one of the most dangerous chemical agents worldwide. Presently, the application of this chemical agent is banned in many geographic areas. Please exemplify other types of modern hazardous agents

Response: Modified.

  • Lines 174-175: which pesticide? Please clearly mention which pesticides increase the prevalence of diabetes.

Response: Modified.

  • Please also describe the occurrence of which diabetes subtype is associated with exposure to pesticides.

Response: Modified.

  • Please mention how pesticides dysregulate gene expression profile and level of transcription factors in the human body. There are several studies in this line that have not been discussed in the paper. For instance, you can see the following references:
  • Int J Environ Health Res: 2021 Nov;31(7):805-822. DOI: 10.1080/09603123.2019.1690132

Answer: This reference has been added and check it is mentioned as Reference 16 in the references.

  • https://doi.org/10.1016/B978-0-323-85160-2.00005-6

Answer: This is the latest book chapter of the book, which has been recently published in 2022. We are sorry to say that we do not have access to this chapter as well as book.

  • https://doi.org/10.1016/j.etap.2018.08.018

Answer: This reference has been added and check it is mentioned as Reference 17 in the references.

  • Please add a table to the text to compare the effectiveness of chemical agents and plant extracts for the preservation of stored fruits and vegetables.

Response: With due respect, this is already discussed in manuscript as well as in Figure

  • Please also discuss the possible effects of the combination of safe nanoparticles with plant extracts to preserve fruits and vegetables.

Response: This manuscript mainly focused on natural plant extract (Aloe vera, lemongrass and neem) for preservation of fruits and vegetables.

  • According to your searches, essential oils are more effective for the preservation of fruits or raw extracts? Preparing plant extracts from contaminated sources with mycotoxins and other biological toxins might increase further concerns about the safety of these products. Please add a short explanation about this case to the paper.

Response: Thanks  a lot for your comment. It is understood that the plant extracts must be collected from healthy parts of plants either oil or raw extract. It is early to say that which one is best oil or raw material. There must be separate research on it that explain the reasons of uses of Oils, raw extracts/material and combinations of extracts or their byproducts to use in food preservation.

  • Plant extracts have some special secondary products that can be easily oxidized under environmental temperature. How this disadvantage of plant-based preservative agents should be limited in storage sites?

Response: We know that all the storage facilities are specific and proper conditions according to food and products. So, there must be proper temperature according to plant extracts we are using for the preservation etc.

  • Due to their natural edible properties, plant extracts can be used in fruit and vegetable preservation by dipping, fumigating, spraying, or by combining a composite coating with a carrier such as cling paper, and the effect is significant. Please discuss these methods regarding the application of plant extracts for preserving fruits and vegetables. Please also discuss which method is more effective compared to others for the protection of stored fruits and vegetables.
  • Please also add a section to discuss the limitation of plant extracts for the protection of fruits. The combinatorial methods for preserving fruits and vegetables from post-harvest pests and diseases have not been discussed in this paper. please add some lines about the possible combination of chemical and plant extracts for preserving stored fruits and vegetables.

Response: It is very nice comment and idea. There must be separate review/research paper on it that explain the methods of application of plant extracts and their advantages and disadvantages. Also, discuss the best one among them.

  • One of the most important complications of plant extracts for preserving stored fruits and vegetables is their effects on the organoleptic properties of foods. A majority of plant extracts have chemical metabolites that can interact with biological components of fruits and vegetable outer peel and might reduce the quality of these products. How this limiting factor should be eliminated in the fruit industry when plant extracts will be used as protective agents? Please briefly discuss this case at the end of the review. The following reference might support the respected authors to discuss this issue:

o   https://www.frontiersin.org/articles/10.3389/fmicb.2021.753518/full

Answer: This reference has been added and check it is mentioned as Reference 121 and 122 in the references.

  • Although plant extracts are safe and edible products (in 98% of all cases), however, there is an import issue associated with these natural products that influenced their large-scale application in fruits and vegetable industry as protective agents. This issue is the stability of plant extracts. In comparison to presently recruited protectant chemicals, plant extracts are less stable and after administration to stored sites they might loss a significant proportion of their biological activity. Please kindly consider some references in the literature to discuss this case in your review.

Response: Dear reviewer, we are not discussing the chemical and biochemical properties of plant extracts in this review. We think that there is no need to discuss that matter here. As an expert researcher, you know that we cannot cover every aspect in one review otherwise it will look like a general review which will cover whole data of the world. Reviews are mainly focused on one aspect in a critical and comprehensive way. 

  • Please check the number and caption of inserted figures. The last two figures’ data can be combined.

Response: Modified.

Overall, this paper provides a novel perspective on the application of plant extracts to preserve fruits and vegetables. It is well-designed and has a good structure. However, considering the above-mentioned comments before publication will boost the quality of the paper, and updating its content increase its impact on scientific readers of this journal.

Round 2

Reviewer 1 Report

Dear Authors, thank you for your response, I have no further comments or suggestions for the edited version of the manuscript.

Reviewer 2 Report

In figure 1.

Change trituration for grinding; Filteratrion for Filtration and coating solution for spraying solution

Reviewer 3 Report

Thanks for providing the answers in detail. I have no further comments on this paper, and it is suitable for publication. 

Regards, 

Rasouli. H